# A Hybrid Millimeter-Wave and Free-Space-Optics Communication Architecture with Adaptive Diversity Combining and HARQ Techniques

Yinjun Liu [1,2], Xiaochuan Tan [3], Junlian Jia [1,2], Boyu Dong [1,2], Changle Huang [1,2], Penghao Luo [1,2], Jianyang Shi [1,2], Nan Chi [1,2] and Junwen Zhang [1,2],*

1 Key Lab of EMW Information (MoE), Fudan University, Shanghai 200433, China; 23110720080@m.fudan.edu.cn (Y.L.); jy_shi@fudan.edu.cn (J.S.); nanchi@fudan.edu.cn (N.C.)
2 Shanghai ERC of LEO Satellite Communication and Applications, Shanghai CIC of LEO Satellite Communication Technology, Fudan University, Shanghai 200433, China
3 The 54th Research Institute of CETC, Shijiazhuang 050081, China; tanxiaochuan333@163.com
* Correspondence: junwenzhang@fudan.edu.cn

**Abstract:** We propose and demonstrate a hybrid communication architecture that combines millimeter-wave (MMW) in the radio frequency (RF) domain and free-space-optics (FSO) technologies using adaptive combining and hybrid automatic repeat request (HARQ) techniques. At the receiving end, we employed joint signal processing with an adaptive diversity combining technique (ADCT) based on a maximum ratio combining (MRC) algorithm. We derived closed-form expressions for the outage probability and throughput of the hybrid RF and FSO (RF/FSO) system, considering various characteristics of atmospheric turbulence in the FSO link. Experimental testing with 10-Gbaud quadrature phase shift keying (QPSK) data was conducted under different simulated atmospheric turbulence intensities, FSO and MMW speed-ratios, and forward error correction (FEC) overheads. Additionally, we validated improvements in terms of bit error ratio (BER), outage probability, and throughput performance.

**Keywords:** free-space-optics (FSO) communications; millimeter-wave (MMW) communications; maximum ratio combining (MRC); hybrid automatic repeat request (HARQ)





## 1. Introduction

The free-space optical (FSO) communication system has emerged as a promising solution for achieving high-speed data transmission over long distances, thanks to its advantages such as large communication capacity, high transmission rate, abundant spectrum resources, and flexible installation. It also has great potential in data communications between the Earth and satellites, as well as inter-satellite links [1]. Millimeter-wave (MMW) communication in the radio frequency (RF) domain shares similar features with FSO communication and has been receiving considerable research attention [2]. However, both FSO and MMW communications are limited to short transmission distance due to high propagation loss and impairments from certain weather conditions [1–6]. The concept of a hybrid RF/FSO system emerged from the recognition that these two links have complementary characteristics. RF signals are susceptible to the influence of rain but immune to fog and clouds, while FSO signals exhibit the opposite behavior. By combining both RF and FSO links, the hybrid system aims to leverage the strengths of each technology to enhance overall system reliability and performance [2–5].

The hybrid RF/FSO system is often implemented as a simultaneous transmission system or switchover systems. The switch over system mainly has two categories: one is the hard-switching system, where only one of the FSO and RF links is used as the data transmission link, with the other as the backup link [4,5]. However, it can only use

one link to transmit data, thus wasting the bandwidth of the other link, reducing the transmission efficiency and the throughput of the system in engineering applications. The other is the soft switching depending on channel hybrid coding and modulation with the simultaneous transmission of the two channels and combined using the maximal ratio combining (MRC) algorithm [6]. The main disadvantage lies in that it does not improve the overall performance of the system when both links are unavailable [7]. Hybrid switching is proposed in [8], where the FSO link is the primary information transmission link. When the received SNR of the FSO link falls below a threshold, the RF link is activated to transmit the same data at the same rate as the FSO link. At the receiver, the transmitted data are obtained using the maximum ratio combining (MRC) algorithm. Both the hard and soft switch system require channel state information (CSI), which brings time delay and data rate loss [9], as well as complicated practical devices [10]. The simultaneous transmission system, however, addresses the over-switching issue and uses adaptive diversity combining technique (ADCT) at the receiver [11]. The CSI is no longer needed, resulting in a more simpler hybrid system compared to switchover systems.

The hybrid automatic repeat request (HARQ) protocols has also been implemented to improve communication quality in satellite–terrestrial transmissions [12]. Works have been done concerning HARQ-aided power optimization for low-earth orbit (LEO) satellite-based FSO systems [13] and incremental redundancy (IR) HARQ-aided LEO satellite-based FSO systems [14,15].

We aimed to find a way to combine the advantages of HARQ technology with the hybrid RF/FSO simultaneous transmission system to further enhance the reliability of satellite-to-ground communication. Therefore, in this paper, we present a novel communication system that leverages the hybrid RF and FSO architecture, incorporating adaptive combining and hybrid automatic repeat request (HARQ) techniques. Figure 1 shows the proposed hybrid RF/FSO architecture for communication between the Earth and satellites. We used FSO as the primary link. When the feedback link conveys CSI, the FSO link continues to transmit the next frame of data, while the RF link retransmits data that failed data validation. This approach addresses the latency introduced by retransmissions and leverages the high stability of the RF link while maintaining high data transmission speed and throughput with FSO. The outage probability is also reduced. We provide theoretical analysis and experimental demonstrations to validate the effectiveness of our approach. We derive closed-form expressions for the outage probability and throughput of the RF/FSO system, considering the characteristics of atmospheric turbulence in the FSO link. Experimental testing with 10-Gbaud QPSK data was conducted under different simulated atmospheric turbulence intensities, FSO and MMW speed-ratio, and FEC overheads. In addition, we also validate improvements in terms of bit error ratio (BER), outage probability, and throughput performance.

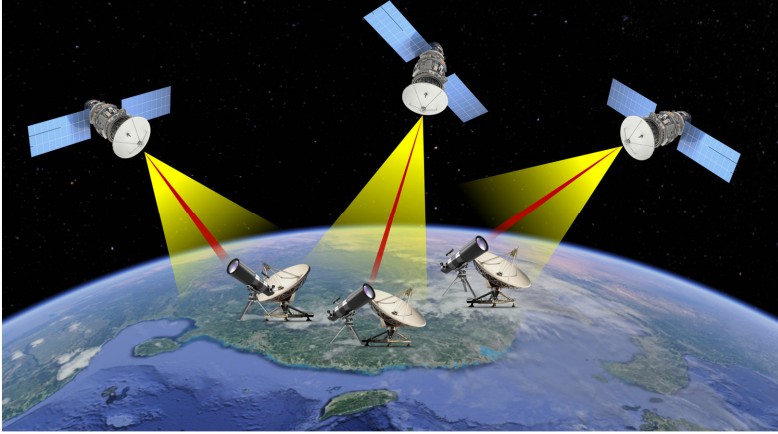

**Figure 1.** Proposed hybrid RF/FSO architecture with ADCT and HARQ for communication between the Earth and satellites [1,3,14].

## 2. Principles

Figure 2 is the architecture of HARQ-aided adaptive diversity combining for hybrid MMW and FSO links system based on the maximum ratio combining (MRC) algorithm. The data to be transmitted is first CRC and FEC encoded and then modulated to achieve a higher transmission rate. The modulated signal sequence that is denoted by $S_{FSO}$ is then n times downsampled at different positions to obtain signal sequences $S_{RF,1}$, $S_{RF,2}$, to $S_{RF,M}$, where $M$ stands for the number of retransmission rounds. The FSO and RF signals are then modulated, respectively, on the FSO laser source and RF source to send corresponding signals. At the receiving side, the signals go through an analog-to-digital convert (ADC) process and carry out traditional communication digital signal processing to obtain two signal sequences, $S'_{FSO}$ and $S'_{RF,M}$, corresponding to FSO link and RF link, respectively. The processed RF sequence $S'_{RF,M}$ and its corresponding position of the FSO sequence $S'_{FSO}$ are then processed by the MRC algorithm. Due to the n times downsampling, $S'_{FSO}$ corresponds to $S'_{RF,M}$ with n symbols at every interval, and the corresponding sampling sequence is denoted as $S_{FSO-down}$. The signal sequence after MRC is denoted as $S_{MRC}$. The rules of MRC are as follows [11]: Assume that the normalized combining gain coefficients of RF and FSO links are set as $\alpha$ (alpha) and $1 - \alpha$, respectively, and then the combined sequence can be expressed as

$$S_{MRC} = \alpha S'_{RF,M} + (1 - \alpha)S_{FSO-down} \tag{1}$$

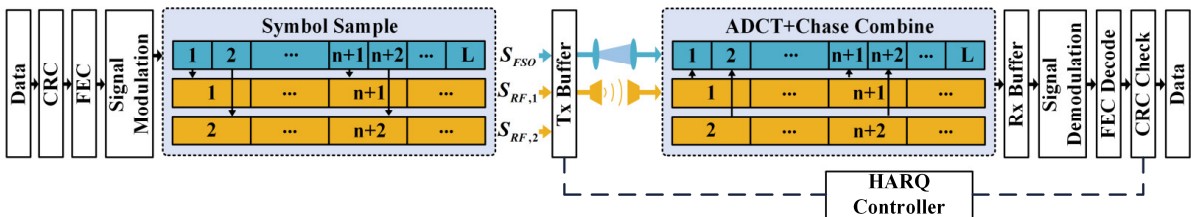

**Figure 2.** Architecture of HARQ-aided adaptive diversity combining for hybrid MMW and FSO link system based on MRC.

According to MRC, by selecting the optimal combining gain coefficient of each channel, the signal-to-noise ratio (SNR) of the combined output signal should be the sum of the SNRs of each sub-link. The normalized SNR of each signal sequence is denoted as $\gamma_{RF}$ and $\gamma_{FSO}$. Hence, we chose $\alpha$, satisfying [8]

$$\alpha = \sqrt{\gamma_{RF}}/(\sqrt{\gamma_{RF}} + \sqrt{\gamma_{FSO}}) \tag{2}$$

We used the combined sequence $S_{MRC}$ to replace the corresponding positions in $S'_{FSO}$; therefore, the downsampled part of the sequence $S'_{FSO}$ will obtain the gain provided by signals transmitted through the RF link and enhance the overall SNR of the received signal. After MRC, $S'_{FSO}$ is demodulated and then decoded by FEC to obtain stronger encoding gain. The power gain of MRC and the coding gain of FEC are used to improve the SNR of the FSO link and reduce the outage threshold of the FSO link, thus improving transmission reliability and ensuring high-speed transmission.

After the FEC decoding, if the CRC check fails, the hybrid automatic repeat request (HARQ) technique intervenes, and the HARQ controller implements RF or RF/FSO hybrid retransmission.

Hybrid automatic repeat request (HARQ) transmission is an error control method that combines forward error correction and automatic request retransmission [16]. HARQ can not only detect error data in the transmission process, but also has certain error correction capability, and thus it can greatly improve system transmission efficiency and reliability. The principle of HARQ technology is shown in Figure 3. We employed low-density parity-check (LDPC) coding in the HARQ protocol [17]. After the sender performs CRC and FEC

encoding, the receiver performs error correction and decoding after receiving the data and then verifies the corrected data. If the data verification of CRC check passes, the decoding is judged to be successful and it sends an ACK signal back to the sender. The sender will continue to send the next set of data. Otherwise, if the data verification fails, the decoding fails. The receiving end will choose to discard or save the received data according to the set rules and send back the NACK signal to the sender to request retransmission until meeting the maximum allowable retransmission times set by the system or the data is received correctly. Therefore, HARQ technology combines the characteristics of FEC and ARQ technology, improves the decoding accuracy, and improves the overall transmission performance of the system effectively. In the type of chase combining HARQ (cc-HARQ), each retransmission block is identical to the original data [18]. The complexity of cc-HARQ is relatively low, and since each retransmission is identical, it is easy for us to combine it with other techniques, in this case, hybrid RF/FSO transmission using ADCT techniques.

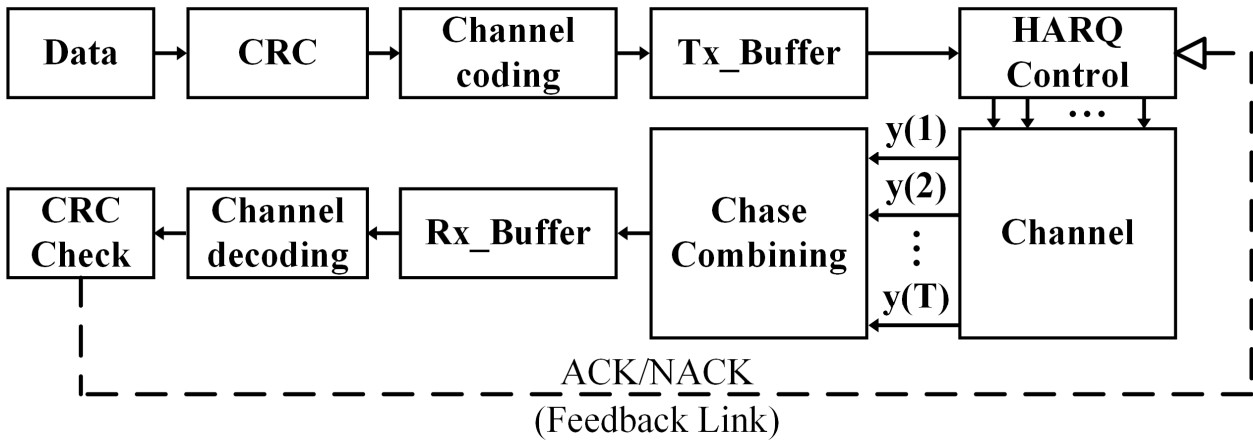

**Figure 3.** Block diagram of the cc-HARQ scheme.

In order to make full use of the time division gain of HARQ under an atmospheric turbulence condition, it is necessary to study the performance of HARQ-aided hybrid FSO/RF communication systems from the perspective of information theory. Based on this, the outage and throughput performance of the system is discussed as follows.

The instantaneous states of the RF link fading coefficient and the FSO link turbulence coefficient at the $i - th$ time slot are denoted as $H_{RF,i}$ and $H_{FSO,i}$, respectively, being referred to as channel coefficients. Assume that these channel coefficients are known at the receiver. In addition, the channel gains are defined as $G_{RF,i} = |H_{RF,i}|^2$, $G_{FSO,i} = |H_{FSO,i}|^2$. It is assumed that the returned channel state information contains only HARQ feedback bits, and the feedback channel can be an RF, an FSO, or an FSO/RF link, being error-free and delay-free. Finally, we assumed that all the links are perfectly synchronized.

For the FSO link, we considered the influence of turbulence in a large range and assumed it follows the Gamma-Gamma fading distribution in strong turbulence conditions and the Log-Normal fading distribution in weak turbulence [19]. The strong and weak turbulence are determined by refractive index structure parameter $C_n^2$, which defines the strength of the refractive index fluctuations in the atmosphere and determines some other important parameters of atmospheric turbulence, such as the scintillation coefficient and Rytov variance. Thus, the probability density function(pdf) of the channel gain is given as follows:

$$f_{G_{FSO}}(x) = \begin{cases} \frac{1}{\sqrt{2\pi}\sigma_I x} \exp\left(-\frac{[lnx+0.5\sigma_I^2]^2}{2\sigma_I^2}\right), & C_n^2 < 10^{-15} \\ \frac{2}{\Gamma(\alpha)\Gamma(\beta)x}\left(\frac{\alpha\beta}{\mu}x\right)x^{\frac{\alpha+\beta}{2}}K_{\alpha-\beta}\left(2\sqrt{\frac{\alpha\beta}{\mu}x}\right), & C_n^2 > 10^{-15} \end{cases} \tag{3}$$

where $f_{G_{FSO}}(x)$ represents the probability density function of the *FSO* channel gain. When $C_n^2 < 10^{-15}$, we use the Log-Normal fading distribution, where $x$ is the received light intensity, which is equivalent to the optical power in unit area, and $\sigma_I$ is the scintillation index. When $C_n^2 > 10^{-15}$, we use the Gamma-Gamma fading distribution, where $K_v(\cdot)$ is the modified Bessel function of the second order. $\mu = E[x]$, and $E[\cdot]$ represents the expected operator. We assumed that the FSO channel gain is normalized, e.g., $\mu = 1$. $\alpha$ and $\beta$ are the fading/scintillation parameters related to the atmospheric turbulence conditions, which can be expressed as functions of Rytov variance $\sigma_R^2 = 1.23C_n^2 k^{7/6}L^{11/6}$, $k = 2\pi/\lambda$ being the wavenumber for wavelength $\lambda$ and $L$ being the transmission distance. The weak, moderate, and strong turbulence conditions are characterized by $C_n^2 = 10^{-15}$, $C_n^2 = 10^{-14}$, and $C_n^2 = 10^{-13}$, respectively [20]. According to the probability density function of the channel gain above, 200 turbulence coefficients of different turbulence intensities are generated at each specific distance, which can be regarded as different turbulence experienced in 200 time slots at the certain distance and turbulence intensity. Figure 4 illustrates the magnitude and distribution of these turbulence coefficients.

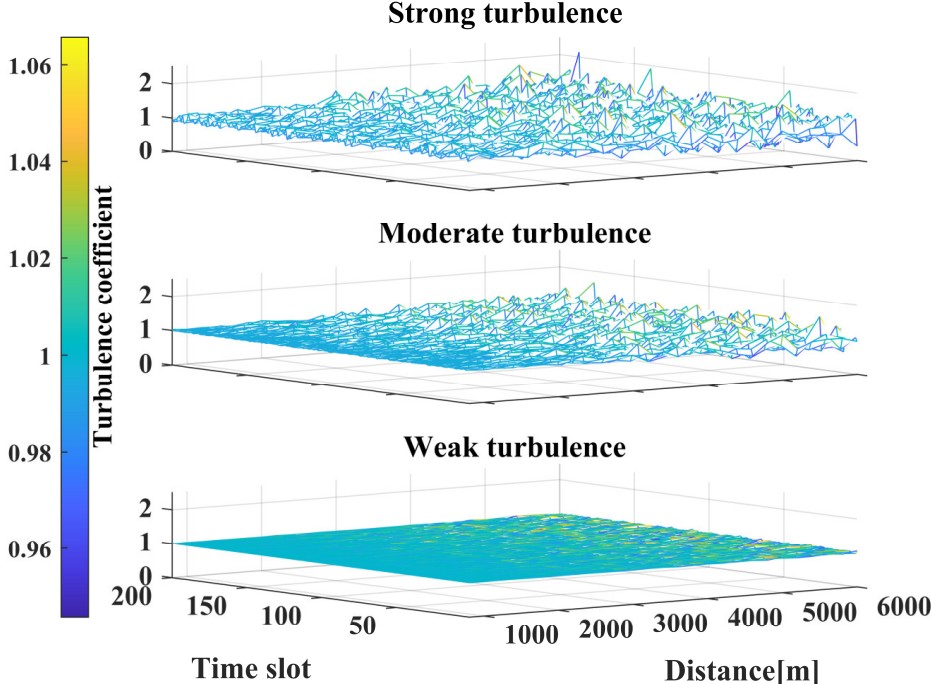

**Figure 4.** The magnitude and distribution of turbulence coefficients in 200 time slots at the certain distance and turbulence intensity.

In FSO/RF systems, the RF link changes very slowly, and the coherence time is $10^2$–$10^3$ times larger than that of the FSO link. For this reason, we can assume the RF link as quasi-static, and its channel remains unchanged during all the time slots [21]. Moreover, during all our simulations and experiments, we maintained the performance of the RF link better even than FSO performance at its best, and the devices of the RF link are all ideal. Consequently, we assumed that the channel gain of the RF link remains 1 during all the transmission and retransmission time slots, i.e., $G_{RF,i} \equiv 1$, which is illustrated in Figure 5. As for the FSO link, different channel coefficient and channel gains are experienced due to the influence of atmospheric turbulence. Every FEC frame contains $8 \times 10^4$ symbols and lasts for 8 μ$s$ at the rate of 10 Gbaud/s. Thus, each transmission and retransmission time slot of the FSO link is $10^3$ times larger than the turbulence variation period which is millisecond order [19]. For this reason, we can assume that during each FSO transmission time slot, the channel remains unchanged. While in each retransmission round, different channel realizations are experienced, as is shown in Figure 5.

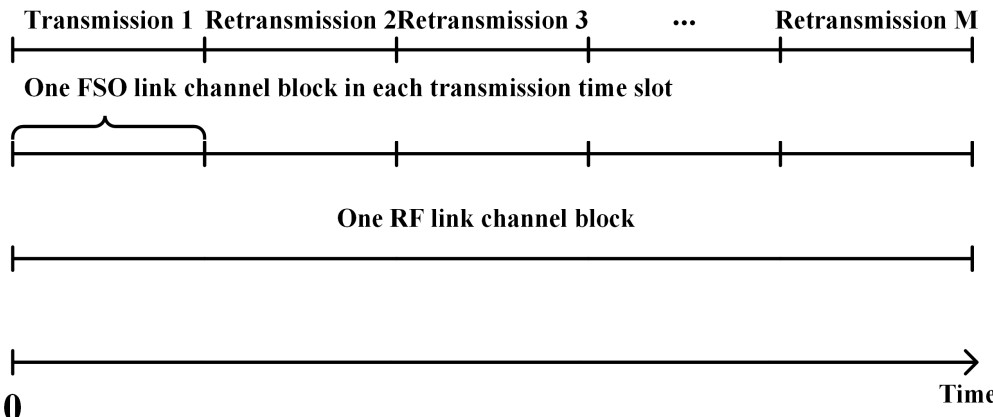

**Figure 5.** The RF link remains constant in the retransmissions, while in each retransmission HARQ round, different channel realizations are experienced in the FSO link.

In this paper, we considered HARQ with chase combining (cc-HARQ), which uses maximum ratio combining (MRC) to store previously failed packets so that they can be combined with subsequent packets. The mutual information accumulated by FSO link after $M_{FSO}$ cc-HARQ rounds is given by [22]

$$I_{FSO}^{CC} = log_2\left(1 + \sum_{k=1}^{M_{FSO}} P_{FSO}G_{FSO}\right) \tag{4}$$

where we denote the signal-to-noise ratio (SNR) of the receiving end of the *FSO* link as $P_{FSO}$, and $P_{FSO}G_{FSO}$ represents the SNR at the receiving end of the *FSO* link under the influence of turbulence. Similarly, considering that the RF channel model gain equals to 1, the accumulated mutual information of RF link after $M_{RF}$ HARQ rounds is

$$\begin{aligned} I_{RF}^{CC} &= log_2\left(1 + \sum_{k=1}^{M_{RF}} P_{RF}G_{RF}\right) \\ &= log_2(1 + M_{RF}P_{RF}) \end{aligned} \tag{5}$$

Considering the difference in transmission rate between RF and FSO links, the above equation becomes

$$I_{FSO}^{CC} = \Psi log_2(1 + M_{RF}P_{RF}) \tag{6}$$

where $\Psi$ represents the relative symbol rates of the RF link over the *FSO* link [21]. It can be determined by the actual scenario. The outage probability is defined as the probability of the event that the accumulated mutual information of the two links combined is less than the transmission rate R [22], which is

$$P_{out} = Pr\{I < R\} \tag{7}$$

where the transmission rate R can be represented by the spectral efficiency (bps/Hz). The outage probability expression of the system is obtained by substituting the two mutual information formulas and considering the pdf of the channel gain

$$\begin{aligned} P_{out} &= Pr\left\{log_2\left(1 + \sum_{k=1}^{M_{FSO}} P_{FSO}G_{FSO}\right) + \Psi log_2(1 + M_{RF}P_{RF}) \leq R\right\} \\ &= Pr\left\{\sum_{k=1}^{M_{FSO}} G_{FSO} \leq \frac{2^{[R-\Psi log_2(1+M_{RF}P_{RF})]}-1}{P_{FSO}}\right\} \end{aligned} \tag{8}$$

When $M_{FSO} = 1$, that is, the FSO link transmits only once, the above formula can be simplified to

$$
\begin{aligned}
P_{out} &= Pr\left\{ G_{FSO} \leq \frac{2^{[R-\Psi log_2(1+M_{RF}P_{RF})]}-1}{P_{FSO}} \right\} \\
&= \int_0^d \frac{2}{\Gamma(\alpha)\Gamma(\beta)x} \left( \frac{\alpha\beta}{\mu}x \right)^{\frac{\alpha+\beta}{2}} K_{\alpha-\beta}\left( 2\sqrt{\frac{\alpha\beta}{\mu}x} \right) dx
\end{aligned}
\tag{9}
$$

where $d = \left[ 2^{[R-\Psi log_2(1+M_{RF}P_{RF})]} - 1 \right] / P_{FSO}$. Rewrite the modified Bessel function in terms of the Meijer-G function [23] (p. 665), and we can obtain

$$
P_{out} = \int_0^d \frac{2}{\Gamma(\alpha)\Gamma(\beta)x} \left( \frac{\alpha\beta}{\mu}x \right)^{\frac{\alpha+\beta}{2}} \frac{1}{2} G_{02}^{20} \left( \frac{\alpha\beta}{\mu}x \Big|_{(\alpha-\beta)/2, -(\alpha-\beta)/2} \right) dx
\tag{10}
$$

where $G(\cdot)$ is the Meijer-G function. Using the integral formula of the Meijer-G function [23] (p. 46), a closed-form expression of the outage probability of cc-HARQ-aided FSO-RF hybrid communications can be expressed as follows:

$$
P_{out} = \frac{1}{\Gamma(\alpha)\Gamma(\beta)x} \left( \frac{\alpha\beta}{\mu}x \right)^{\frac{\alpha+\beta}{2}} \frac{1}{2} G_{13}^{21} \left( \frac{\alpha\beta}{\mu}x \Big|_{-(\alpha-\beta)/2, -(\alpha+\beta)/2}^{1-(\alpha+\beta)/2, (\alpha-\beta)/2} \right) \Big|_{x=d}
\tag{11}
$$

When the messages on the FSO link are transmitted more than once, i.e., $M_{FSO} > 1$, according to the distribution of the sum of k independent identically distributed (i.i.d) random variables with a Gamma-Gamma distribution [22], it is easy to obtain the closed-form expression of outage probability when the FSO link is transmitted for $M_{FSO}$ times and the RF link is transmitted for $M_{RF}$ times

$$
\begin{aligned}
P_{out} &= \frac{1}{\Gamma(\sigma_{M_{FSO}})\Gamma(\zeta_{M_{FSO}})x} \left( \frac{\sigma_{M_{FSO}}\zeta_{M_{FSO}}}{\mu}x \right)^{\frac{\sigma_{M_{FSO}}+\zeta_{M_{FSO}}}{2}} \\
&\times \frac{1}{2} G_{13}^{21} \left( \frac{\sigma_{M_{FSO}}\zeta_{M_{FSO}}}{\mu}x \Big|_{-(\sigma_{M_{FSO}}-\zeta_{M_{FSO}})/2, -(\sigma_{M_{FSO}}+\zeta_{M_{FSO}})/2}^{1-(\sigma_{M_{FSO}}+\zeta_{M_{FSO}})/2, (\sigma_{M_{FSO}}-\zeta_{M_{FSO}})/2} \right) \Big|_{x=d}
\end{aligned}
\tag{12}
$$

where $d = \left[ 2^{[R-\Psi log_2(1+M_{RF}P_{RF})]} - 1 \right] / P_{FSO}$, $\sigma_k$ and $\zeta_k$ are respectively given by

$$
\sigma_k = k\upsilon,
\tag{13}
$$

$$
\zeta_k = k\tau,
\tag{14}
$$

where $\upsilon = \max\{\alpha, \beta\}$, $\tau = \min\{\alpha, \beta\}$. $\mu$ is the mathematical expectation that is equal to 1 if we assume that the FSO channel gain is normalized. $\alpha$ and $\beta$ are the fading/scintillation parameters related to the atmospheric turbulence conditions. These are the distribution parameters of a single variable that follows the Gamma-Gamma fading distribution.

We can also analyze the throughput of the system under different RF over FSO rate ratios and retransmission numbers. The throughput is the average rate of data successfully decoded at the receiving end, and it is given by [21,22]

$$
\eta = \frac{R(1 - p(M))}{1 + \sum_{m=1}^{M-1} p(m)}
\tag{15}
$$

where $R$ is the transmission rate of the first transmission round and $p(m)$ denotes the probability of the receiver failing to correctly decode the message after m rounds. Under the assumption of random coding and typical set decoding with large packet lengths, $p(m)$ is equivalent to the outage probability after m HARQ rounds. By substituting the above outage expression in Equation (15), we can obtain the closed-form approximation for the throughput of cc-HARQ.

### 3. Numerical Simulation

In this section, we illustrate the numerical simulation of outage probability and throughput of the system when facing a strong turbulence condition, based on the principles and modeling proposed on the previous section. In addition, we study the effect of various system parameters on the performance. The RF link receiving side SNR $P_{RF}$, transmission rate R, and relative symbol rate $\Psi$ were all measured in subsequent experiments or adopted as actual values.

Table 1 lists the parameters we used in the simulation. Under the strong turbulence condition, the refractive index structure parameter $C_n^2$ is in the order of $10^{-13}$ [24]. By substituting $C_n^2$, simulation transmission distance L, wavelength $\lambda$ into $\sigma_R^2 = 1.23 C_n^2 (2\pi/\lambda)^{7/6} L^{11/6}$, specific values for the Rytov variance $\sigma_R^2$ can be calculated, as well as the scintillation parameters $\alpha$ and $\beta$. All the simulations are under the conditions that the receiving side SNR of the RF link is fixed as 41.2, which is better than the best of FSO link, and it is a value obtained through experimental measurements. The relative symbol rate of the RF link over the FSO link is 0.25, and the modulation format is QPSK, which means the transmission rate R = 2. The RF link receiving side SNR $P_{RF}$, transmission rate R, and relative symbol rate $\Psi$ were all measured in subsequent experiments or adopted as actual values.

**Table 1.** Simulation parameters.

| Name | Symbol | Value |
|---|---|---|
| Refractive index structure parameter | $C_n^2$ | $10^{-13} \mathrm{m}^{-2/3}$ |
| Simulation transmission distance | L | 2800 m |
| Wavelength | $\lambda$ | 1550 nm |
| Scintillation parameter | $\alpha$ | 10.31 |
| Scintillation parameter | $\beta$ | 37.38 |
| Rytov variance | $\sigma_R^2$ | 13.15 |
| RF link receiving side SNR | $P_{RF}$ | 41.2 dB |
| Transmission rate | R | 2 pbs/Hz |
| Relative symbol rate | $\Psi$ | 0.25 |

It should be noted that, in the previous theoretical analysis such as in Equation (9), $M_{FSO}$ and $M_{RF}$ represent the number of transmissions for the *FSO* and RF links, respectively. If it equals 1, it means one transmission; if it equals 2, it indicates one transmission followed by one retransmission, and so on. The theoretical analysis considered retransmissions separately for the *RF* and *FSO* links to more clearly analyze the impact of various parameters on system performance. However, in the subsequent simulations and experiments, we propose two retransmission strategies: one involves retransmitting only the RF link, and the other involves retransmitting both the RF and FSO links together. In these two retransmission strategies, either only the RF link's retransmissions are considered (in this case, $M = M_{RF}$), or the number of retransmissions for both links is the same (in this case, $M = M_{RF} = M_{FSO}$). Therefore, in representing the number of transmissions, we can omit the subscripts and simply use $M$ to denote it.

Figure 6 illustrates the outage probability and throughput versus the receiving side SNR of the FSO link at different HARQ rounds and HARQ strategies when the FEC overhead equals 5%. It is observed from Figure 6a that although the symbol rate and the transmission rate of the RF link is a quarter to the FSO link, when M = 2 and the RF link conduct a HARQ round, the outage probability still descends significantly compared to the single FSO link. The system performance of the RF link retransmission twice is similar to that of RF and FSO link retransmission once together. In terms of throughput performance, when the SNR on the receiving end of the FSO link is small, the FSO link is required to participate in retransmission to achieve a higher throughput due to the large rate ratio to the RF link. However, when SNR increases, the RF-only retransmission strategy can achieve higher throughput due to the fact that when the RF link is conducting a HARQ round, the FSO link can transmit the next data frame. The throughput of the single RF

link HARQ strategy outperformed the RF/FSO hybrid HARQ strategy in the large SNR condition, as is shown in Figure 6b.

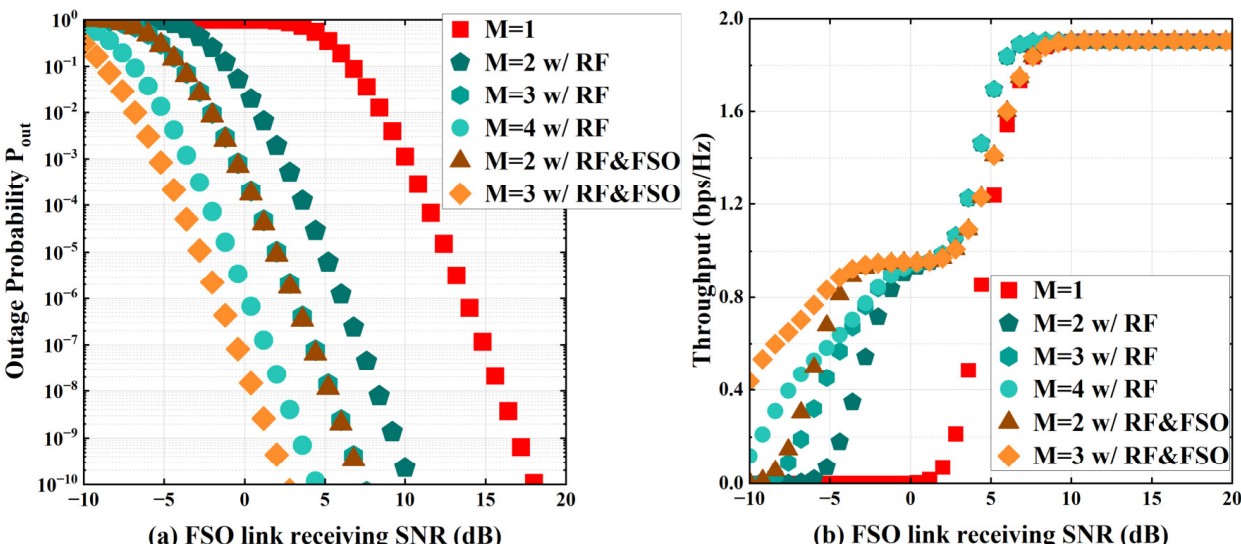

**Figure 6.** The outage probability and throughput versus the receiving side SNR of the FSO link at different HARQ rounds and HARQ strategies when the FEC overhead equals 5%: (**a**) outage probability; (**b**) throughput.

Figure 7 illustrates the outage probability and throughput versus the receiving side SNR of the FSO link at different HARQ rounds and HARQ strategies when the FEC overhead equals 15%. It is observed that the performance of the system when the RF link retransmitted twice was much better than the system performance of the RF/FSO hybrid retransmitting once together compared to the aforementioned OH = 5%. And the outage probability became zero when RF retransmitted three times. The larger the OH, the greater the gain brought by RF retransmission. However, in terms of the throughput, because of the larger proportion of OH occupying the transmitted data frame, the total net amount of information became less and the total throughput decreased compared to the OH = 5% condition.

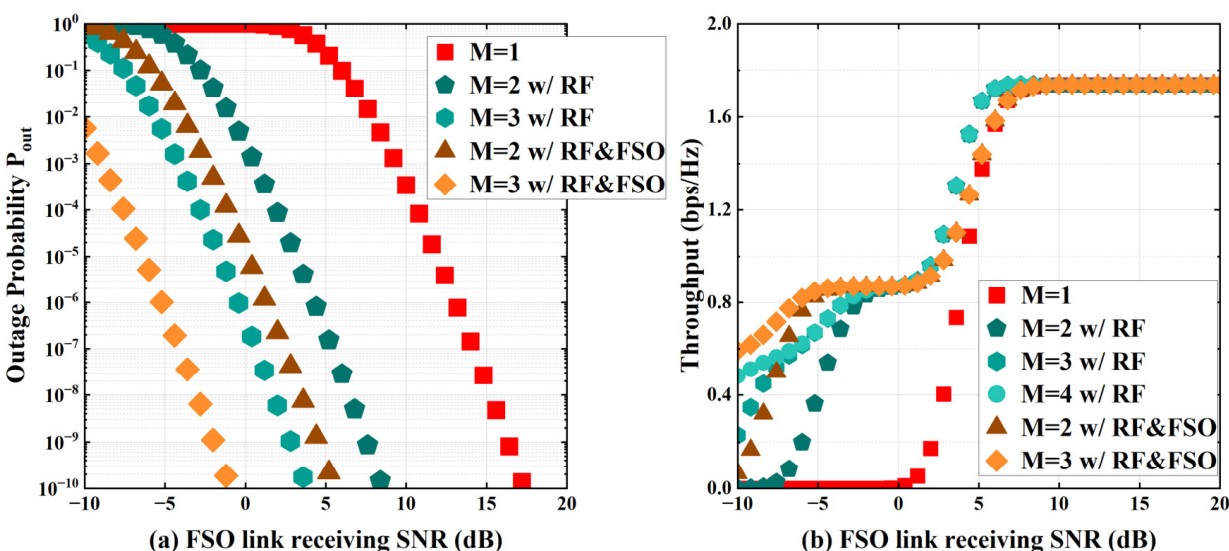

**Figure 7.** The outage probability and throughput versus the receiving side SNR of the FSO link at different HARQ rounds and HARQ strategies when the FEC overhead equals 15%: (**a**) outage probability; (**b**) throughput.

Figure 8a depicts the outage probability versus the receiving side SNR of the FSO link under different FEC OH with RF-only retransmission. Figure 8b depicts the throughput versus the receiving side SNR of the FSO link under the same circumstances. The overhead of FEC affects the total amount of net information transferred. When the length of the data frame to be transmitted is fixed, the change in the net information transmitted due to a different FEC overhead can be equivalent to a change in the modulation order and thus to a change in the transmission rate R. From Figure 8a, it is observed that the higher the FEC OH cost, the lower the transmission rate, the lower the interrupt probability, and therefore system performance is improved. Thus, it can be concluded that when the total number of bits transmitted is constant, the gain of FEC to the system transmission is brought by the coding gain on the one hand and the decrease in equivalent modulation order on the other. As for the throughput, it is shown in Figure 8b that the higher the FEC OH cost, the lower the throughput. This was due to the reduction of net information transmitted when the proportion of OH was higher.

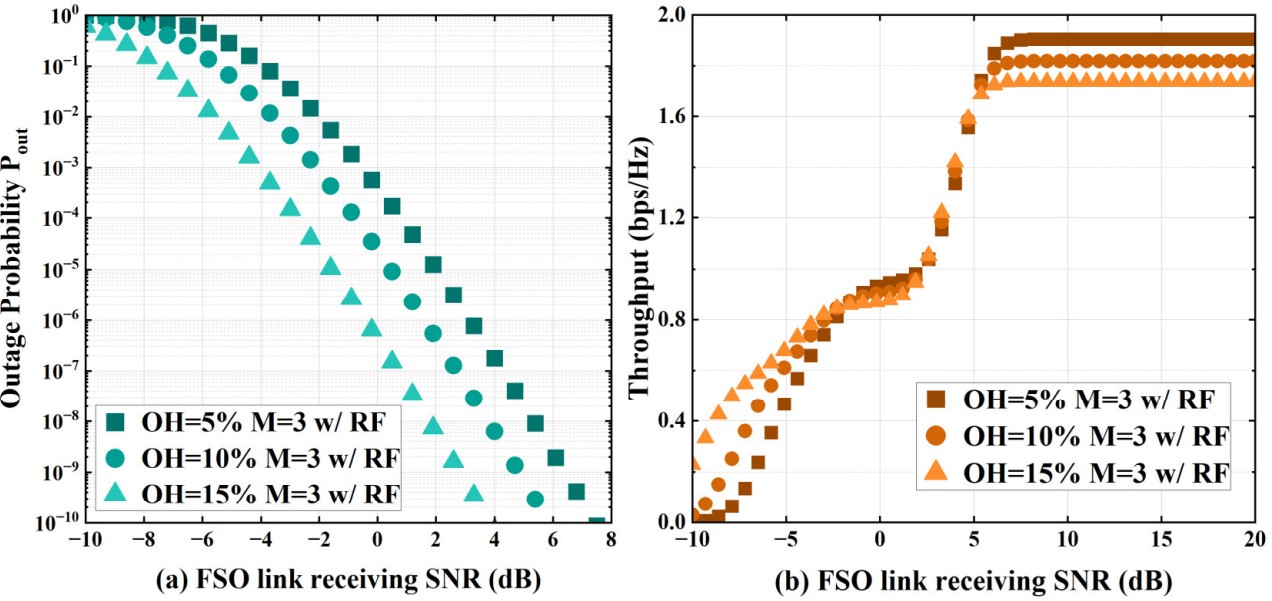

**Figure 8.** The outage probability and throughput versus the receiving side SNR of the FSO link under different FEC overheads with twice RF-only retransmission: (**a**) outage probability; (**b**) throughput.

## 4. Experiment

The experimental setup is shown in Figure 9. In the transmission digital signal processing (Tx DSP) block, the data stream to be transmitted was first CRC encoded, then FEC encoded, and then modulated into QPSK signals, as is introduced in Figure 2. The output modulated symbol stream is denoted as $S_{FSO}$; it is then downsampled N times to obtain bit stream $S_{RF,N}$. In the following experiment, two times and four times downsampling were adopted according to the different transmission rate ratios, that is, the number n in the bit sample block is 2 and 4, respectively. Depending on the retransmission times of the RF link, down-sampling sequences of different positions are generated. Sequence $S_{RF,1}$ and $S_{RF,2}$ in the figure represent the transmitting data stream when RF link is retransmitted once and twice, respectively. After the symbol sample, the signal stream of the FSO link $S_{FSO}$ and the signal stream of the RF link $S_{RF,N}$ were then sent into a 60-GSa/s sampling rate arbitrary waveform generator (AWG). The output electrical signals of the $S_{FSO}$ and $S_{RF,N}$ modulated sequences were used to drive the Machzender modulator (MZM) of the FSO link and RF link, respectively. In the FSO link, Laser 1 operated at 1550 nm and was used as the light source of the FSO link. The output signal light passed through an erbium-doped fiber amplifier (EDFA) and a variable optical attenuator (VOA) before transmission. The VOA was used to adjust the transmitted optical

signal power. A pair of fiber collimators served as antennas in the FSO link. The collimator in the transmission side was used to collimate free space light propagating from the tip of the fiber. After 3.8 m of line-of-sight (LoS) free space transmission, another collimator was used to focus the received FSO light into the fiber at the receiving end. The received FSO signals were then detected by a 10 GHz photodetector (PD) and sent into a digital storage oscilloscope (DSO) with a sampling rate of 100 GHz for subsequent joint off-line digital signal processing (DSP). In the RF link, the 80 GHz MMW signal was generated by the optical beat method. Laser 2 operated at 193.1 THz and was used as the modulated signal source for RF links. Laser 3 operated at 193.18 THz and was used as the local oscillator light of the RF link with an output optical power of 7 dBm and a frequency difference of 80 GHz from Laser 2. It is noted that the power and the polarization of the beat light and the modulated signal light should be consistent so that the generated MMW signal can obtain the optimal heterodyne gain. A 3 dB polarization maintaining optical coupler (PM-OC) was utilized to combine the optical signals and the LO light. The combined optical signal was sent into a high-speed PD with 100-GHz bandwidth and was up-converted to MMW. A pair of Cassegrain antennas with 45-dBi gains was used to transmit the MMW signal.

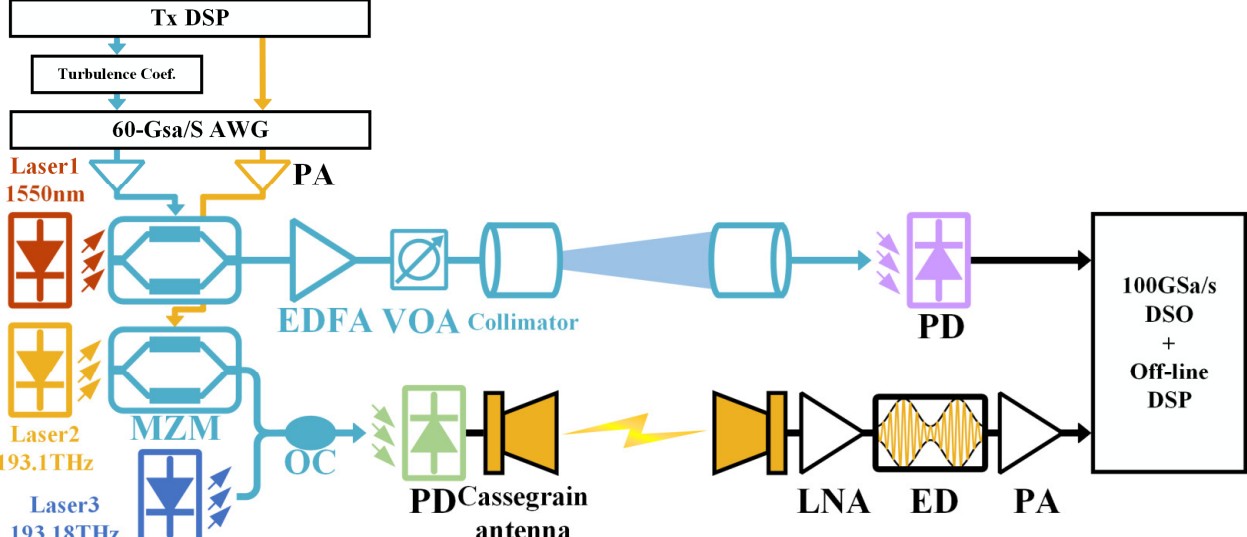

**Figure 9.** Experimental setup of the hybrid MMW and FSO links system with HARQ and adaptive combining techniques (PA: power amplifier; MZM: Mach–Zehnder modulator; EDFA: erbium-doped fiber amplifier; PD: photodetector; ED: envelop detector; DSO: digital storage oscilloscope; LNA: low noise amplifier).

After 4 m LoS transmission, the MMW signal was received by a low noise amplifier (LNA), and then it was converted down to an intermediate frequency (IF) signal by an envelope detector (ED) and amplified by a power amplifier (PA). It was then stored by the DSO for subsequent joint off-line digital signal processing. The offline digital signal processing block is shown in Figure 2, where the signals transmitted and received by MMW and FSO links were synchronized and resampled, respectively, and the channel estimation was also carried out. Then, the two synchronized signals were combined by the ADCT based on the MRC algorithm with the best binding ratio $\alpha$ determined by the SNR of the two channels. If the CRC check failed, then the HARQ controller would implement RF or RF/FSO hybrid retransmission.

We generated a series of quantifiable time-varying turbulence coefficients based on channel and turbulence models. As mentioned earlier, due to the much shorter transmission time of each signal frame compared to the coherence time of turbulence, channel gain was assumed to remain constant during the transmission of each FSO data stream. Therefore, the impact of turbulence on the overall data flow was consistent and the turbulence-induced optical power fluctuations in received power were able to be reflected by applying the

same attenuation coefficient to the signal at the receiving end, and this was equivalent to applying the same attenuation coefficient at the transmitting end. Thus, the influence of atmospheric turbulence on the FSO link was simulated by applying different turbulence coefficients on the data stream of the transmit side, as is shown in Figure 9.

Figure 10 shows the experimental setup of our hybrid RF/FSO transmission system, and Table 2 lists the basic experimental quantities' values. On the left side is the transmitter, where the RF link signal was converted into millimeter-wave signals through laser beating and then passed through a PD before being transmitted simultaneously with the FSO link. The combined optical signal before the PD was 4.43 dBm. At the receiver, the RF link signal was received by an antenna and then went through LNA with the gain of 38 dB, ED, and PA with the gain of 23 dB, before being input into a digital storage oscilloscope. The FSO link, on the other hand, was collimated by a collimator, where the aperture of the FSO antenna used was 11.2 cm. It was then input directly into a PD for photoelectric conversion before being sent to a digital storage oscilloscope, where it awaited further joint digital signal processing.

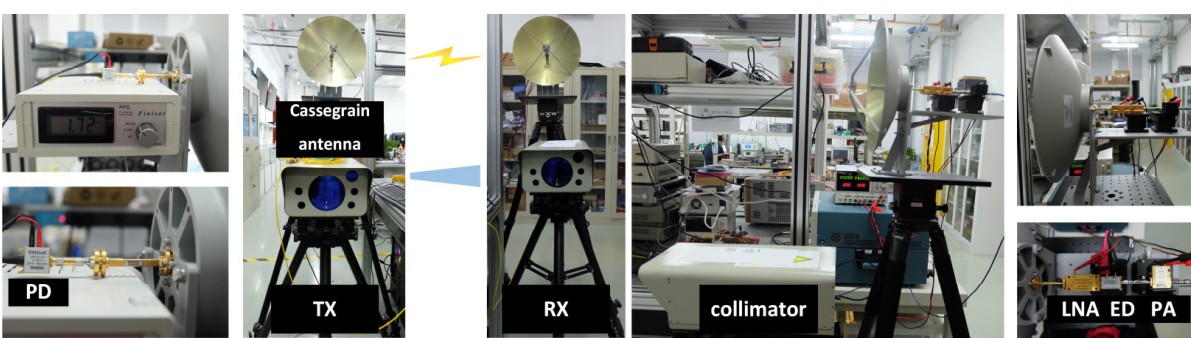

**Figure 10.** Photos of the experimental setup.

**Table 2.** Basic experimental quantities' values.

| Name | Value |
|---|---|
| FSO transmission distance | 3.8 m |
| RF transmission distance | 4 m |
| The gain of PA | 23 dB |
| The gain of LNA | 38 dB |
| The gain of the Cassegrain antenna | 45 dBi |
| The aperture of the FSO antenna | 11.2 cm |

## 5. Experimental Results and Discussion

We first measured the BER performances of different FEC overheads and HARQ strategies under the clean weather condition. Figures 11 and 12 show the BER performances of FSO-only, FSO with FEC, RF-only HARQ once or twice, and RF/FSO hybrid HARQ once or twice versus the transmitted optical power of the FSO link. Figure 11 is under the circumstance that the relative symbol rate of the RF link over the FSO link $\Psi = 0.25$, while Figure 12 is for 0.5.

We can see from Figure 11b that under the $\Psi = 0.25$ condition, when FEC OH = 5%, RF-only retransmission once, twice, and three times obtained 0.4, 0.8, and 1.5 dB power gain, respectively, compared to the FSO-only FEC, while RF/FSO hybrid retransmission once and twice obtained 1.7 and 2.9 dB power gain, respectively. When FEC OH = 15%, the power gain of RF-only retransmission became 0.1, 0.5, and 1.8 dB, and RF/FSO hybrid retransmission became 1.8 and 2.8 dB. The performance of the system for RF/FSO hybrid retransmission once outperformed that of the RF-only retransmission twice. From the perspective of different FEC overheads, when OH = 15%, the performance of RF retransmission once was the same as that of the OH = 5% RF/FSO hybrid retransmission once. Compared with the FEC OH = 5% condition, single FSO transmits without HARQ; RF

retransmits once, twice, and three times; and RF and FSO retransmit once and twice to obtain power gains of 1.6, 1.3, 1.3, 2.1, 1.9, and 1.5 dB. It can be seen that increasing the FEC overhead improved the system performance. However, it should be noted that the increase in the overhead reduced the throughput, and the FSO link was able to continue the transmission of the next data frame with RF-only retransmission when the received SNR of the FSO link was high enough. Consequently, the throughput was much higher than that of the RF/FSO hybrid retransmission, which can also be seen in the previous numerical simulation.

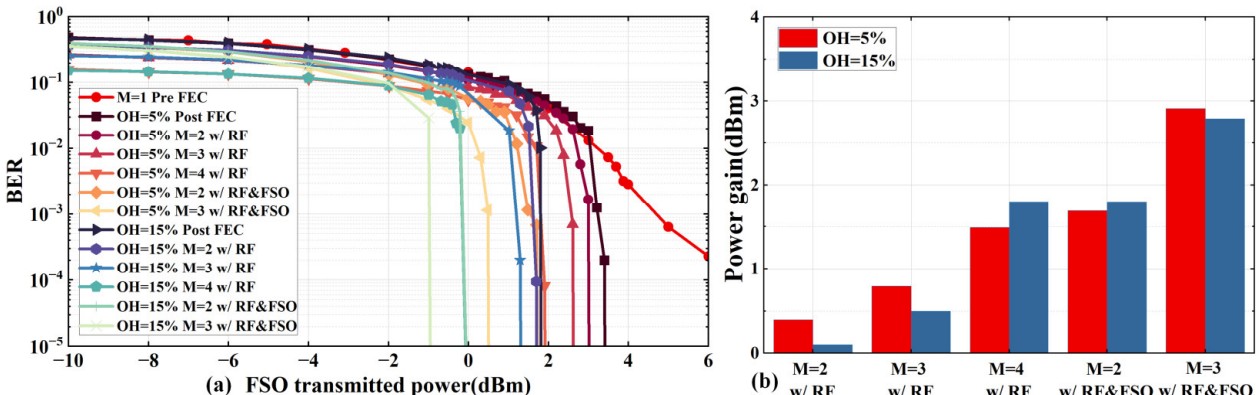

**Figure 11.** The performances of FSO-only, FSO with FEC, RF-only HARQ once or twice, and RF/FSO hybrid HARQ once or twice, $\Psi = 0.25$: (**a**) the BER performances versus the transmitted optical power of the FSO link; (**b**) the power gain of each HARQ strategy of different FEC overheads.

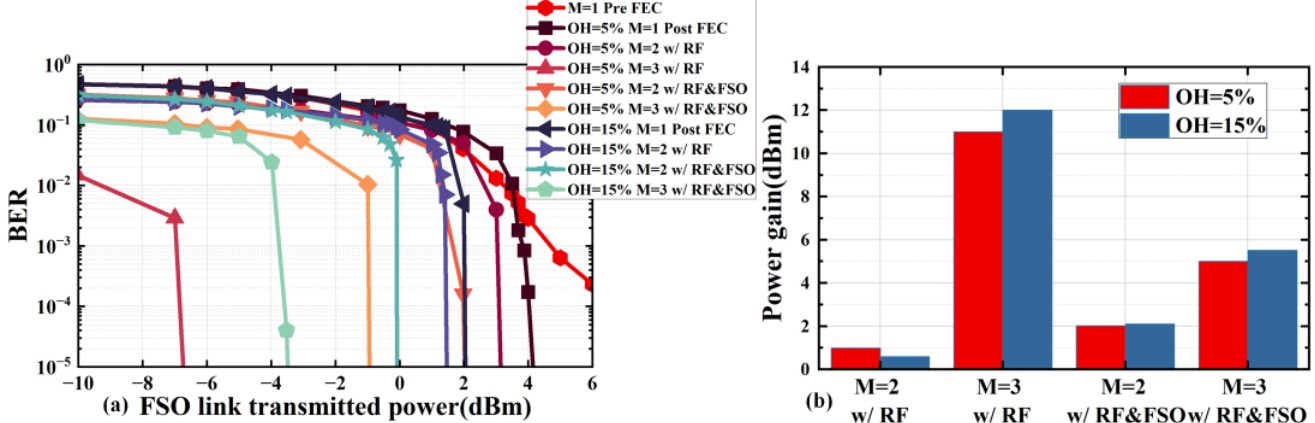

**Figure 12.** The performances of FSO-only, FSO with FEC, RF-only HARQ once or twice with FEC, and RF/FSO hybrid HARQ once or twice with FEC, $\Psi = 0.5$: (**a**) the BER performances versus the transmitted optical power of the FSO link; (**b**) the power gain of each HARQ strategy under different FEC overheads.

As for the $\Psi = 0.5$ condition in Figure 12, when FEC OH = 5%, RF-only retransmission once and twice obtained 1.0 and 11.0 dB power gain, respectively, while the RF/FSO hybrid retransmission once and twice obtained 2.0 and 5.0 dB power gain, respectively. When FEC OH = 15%, RF-only retransmission once and twice obtained 0.6 and 12.0 dB power gain, while RF/FSO hybrid retransmission once and twice obtained 2.1 and 5.5 dB power gain, respectively. Compared with the FEC OH = 5% condition, single FSO transmits without HARQ, RF retransmits once or twice, and RF and FSO retransmit once or twice to obtain power gains of 2.0, 1.6, 3.0, 2.1, and 2.5 dB. RF-only retransmission twice can achieve error-free transmission, even when the FSO transmitted power is −10 dBm and the received power is −27.8 dBm, thanks to the higher transmission rate of the RF link. The RF link transmits half of the information of the FSO link, and thus it achieves larger gain when

retransmitted twice compared to the RF/FSO hybrid retransmission. The FSO signal acts as noise at low received power and influences the MRC process in cc-HARQ, resulting in poor performance of the system with RF/FSO hybrid retransmission twice compared to the system with RF-only retransmission twice.

From the perspective of different relative symbol rates of the RF link over the FSO link Ψ, Figure 13 shows the performances of FSO-only, FSO with FEC, RF-only HARQ once or twice with FEC, and RF/FSO hybrid HARQ once or twice with FEC under different Ψ. We can see from Figure 13e that the first HARQ round achieved similar performance, no matter which HARQ strategy was taken. The second HARQ round showed a great difference in performance. When OH = 5%, Ψ = 0.5 compared with Ψ = 0.25, the power gains of 9.6 and 1.5 dB were obtained under RF retransmission twice and RF/FSO hybrid retransmission twice, and when OH = 15%, the power gains of 11.3 and 2.5 dB were obtained, respectively.

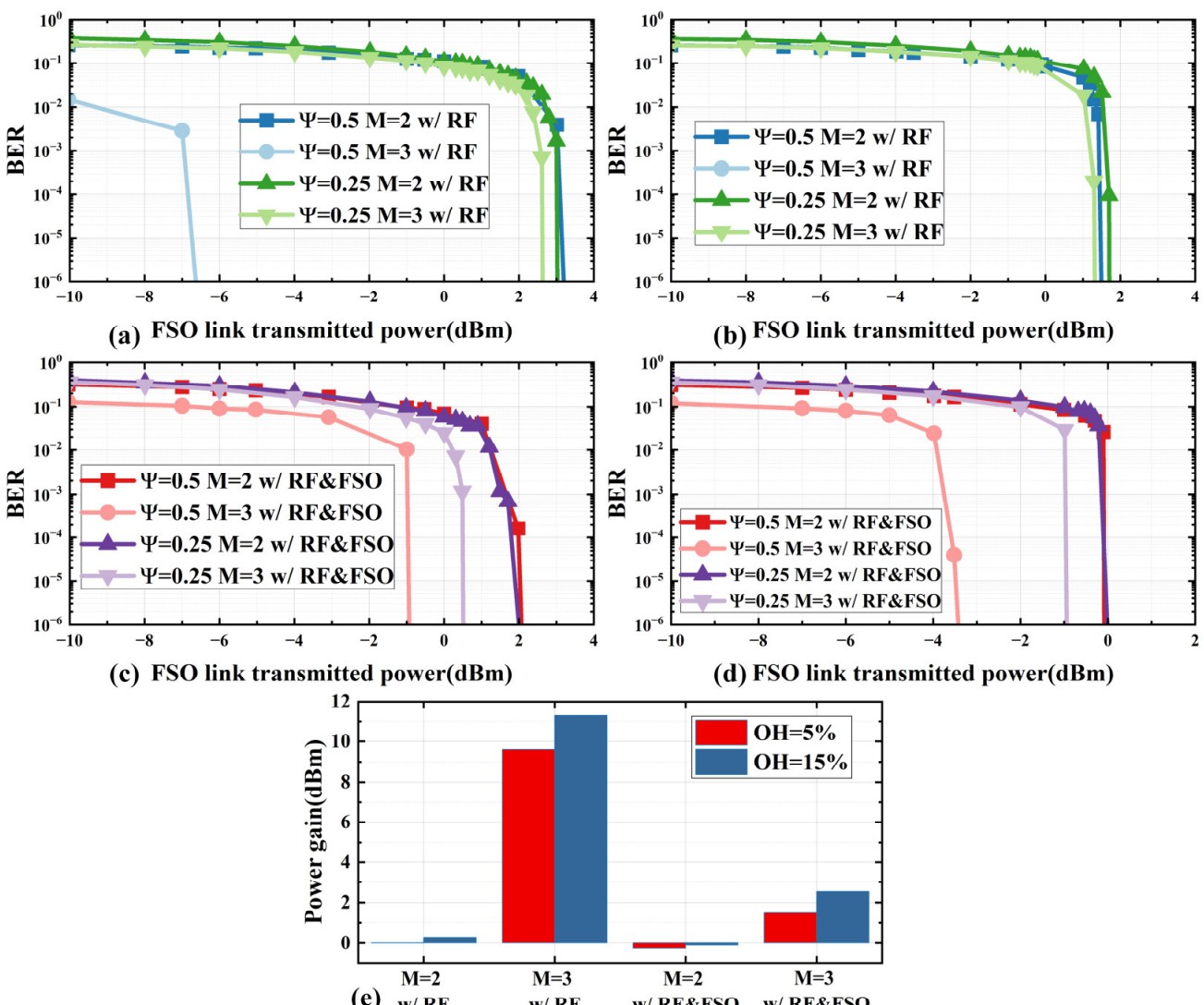

**Figure 13.** The performances of FSO-only, FSO with FEC, RF-only HARQ once or twice, and RF/FSO hybrid HARQ once or twice under different Ψ: (**a**) the BER performances with the RF-only HARQ strategy when FEC OH = 5%; (**b**) the BER performances with the RF-only HARQ strategy when FEC OH = 15%; (**c**) the BER performances with the RF/FSO hybrid HARQ strategy when FEC OH = 5%; (**d**) the BER performances with the RF/FSO hybrid HARQ strategy when FEC OH = 15%; (**e**) the power gain of each HARQ strategy under different FEC overheads.

Next, we probed into the system performance utilizing the HARQ strategy under atmospheric turbulence. The implementation of various air-turbulence simulators makes it difficult to quantify the difference between weak, moderate, and strong turbulence. Since atmospheric turbulence essentially affects the amplitude of the signal being transmitted, we simulated the influence of turbulence on the system by generating the channel gain coefficient under turbulence of different intensities, just as is shown in Figure 4, and then applying the turbulence coefficient directly to the signal at the transmitting end. We chose the turbulence coefficients at 21 time slots appropriately, which contained the worst-case scenario, e.g., the smallest coefficient, of the three turbulence intensities. We examined the BER performances of all 21 time slots at strong, moderate, and weak turbulence intensities; three FEC overheads of 5%, 10%, and 15%; and two RF to FSO data rate ratios $\Psi$ of 1:2 and 1:4.

We first tested the system BER performance under strong atmospheric turbulence. Figure 14a is the strong turbulence coefficients at 21 different time slots, Figure 14b is the pre-FEC BER of FSO link without retransmission, Figure 14c–e is the BER performance at $\Psi = 0.25$ and different FEC overheads, and Figure 14f–h is the $\Psi = 0.5$ condition. This is noted because of the varied channel performances throughout our testing, Figure 14b shows the pre-FEC performance of continuous tests for 21 time slots, and the post-FEC performance differs between Figures 14c–e and 14f–h at the same FEC overhead. When the worst case of strong turbulence was encountered, the turbulence coefficient was 0.58 at the 11th time slot, as is shown in Figure 14a, indicating that the signal amplitude at the receiving end was 0.58 times that without turbulence. When FEC OH = 5%, no matter the RF to FSO data rate $\Psi$ is, 1:2 or 1:4, the system can be transmitted to error-free by RF-only retransmission twice, as can be seen from Figure 14c,f. As is discussed above, the RF-only retransmission strategy at $\Psi = 0.25$ had the worst system performance. Therefore, if the goal was to maintain high throughput in this system at the minimum RF link power, then the error-free transmission would be achieved through two RF retransmissions at strong atmospheric turbulence. When using the RF/FSO hybrid retransmission strategy at FEC OH = 5%, the error-free transmission was achieved through one RF/FSO hybrid retransmissions. We also recorded the system BER performance when FEC OH = 10%, wherein the error-free transmission was achieved through one RF retransmission, as is shown in Figure 14d,g. When FEC OH = 15%, the error-free transmission was achieved right after the FEC decoding, as is shown in Figure 14e,h. Thus, we can draw the conclusion that when the retransmission round was limited, throughput or system consumed power can be sacrificed for the similar BER performance with more retransmission rounds by either utilizing the FSO link to retransmit together, increasing the FEC overhead, or improving the RF data rates.

We then examined the system performances under moderate and weak atmospheric turbulence. Figure 15a is the moderate turbulence coefficients at 21 different time slots, Figure 15b is the pre-FEC BER of the FSO link without retransmission, Figure 15c,d is the BER performance at $\Psi = 0.25$ and different FEC overheads, and Figure 15e,f is the $\Psi = 0.5$ condition. As mentioned above, Figure 15b shows the pre-FEC performance for 21 time slots in one of our selected sets of continuous tests, and the post-FEC performance differed between Figure 15c,d and Figure 15e,f at the same FEC overhead. As is illustrated in Figure 15c–f, the system can achieve error-free transmission by RF-only retransmission once at 5% FEC OH and right after FEC decoding at 10% FEC OH, regardless of the RF to FSO data rate $\Psi$.



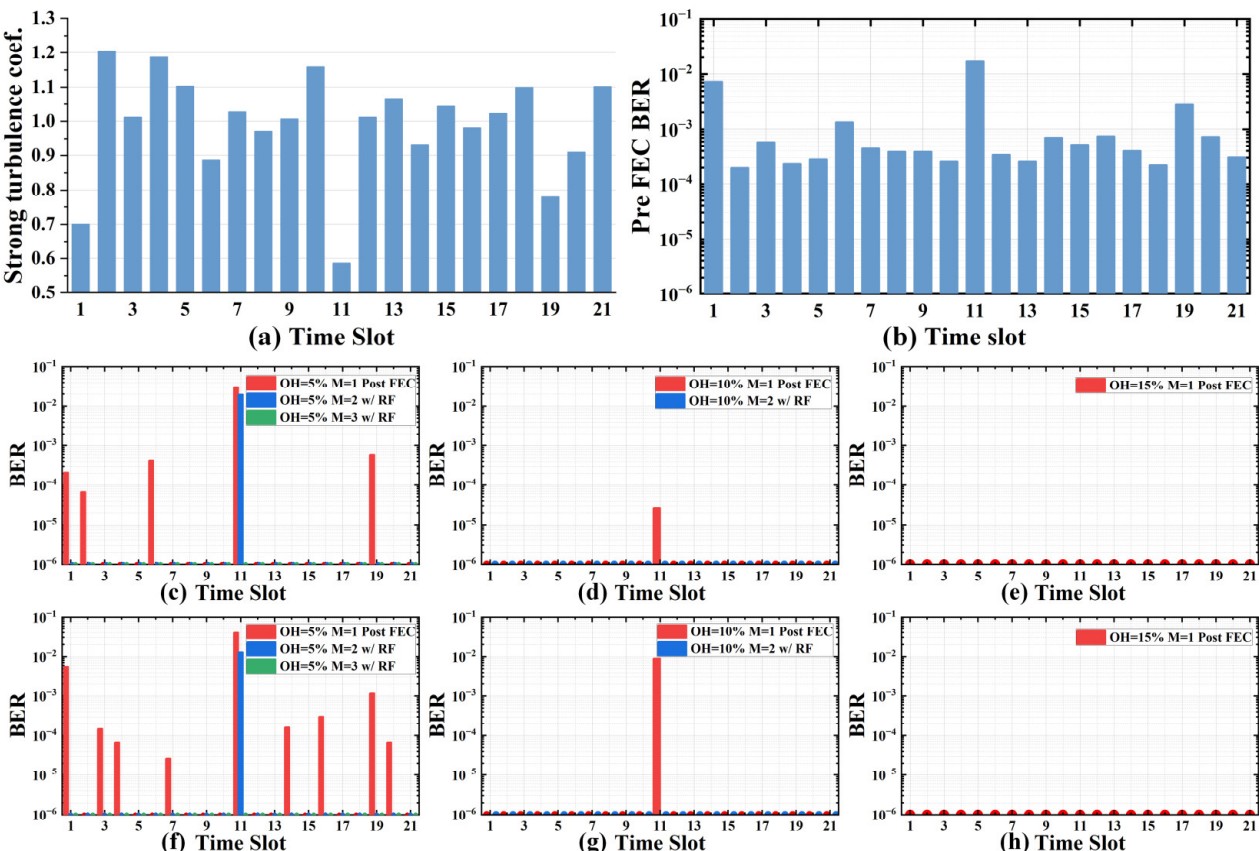

**Figure 14.** The BER performance under strong atmospheric turbulence: (**a**) strong turbulence co-efficients at 21 different time slots; (**b**) the pre-FEC BER of the FSO link without retransmission; (**c**) 5% OH with HARQ-aided BER performances at $\Psi = 0.25$; (**d**) 10% OH with HARQ-aided BER performances at $\Psi = 0.25$; (**e**) 15% OH with HARQ-aided BER performances at $\Psi = 0.25$; (**f**) 5% OH with HARQ-aided BER performances at $\Psi = 0.5$; (**g**) 10% OH with HARQ-aided BER performances at $\Psi = 0.5$; (**h**) 15% OH with HARQ-aided BER performances at $\Psi = 0.5$.

Figure 15g shows the weak turbulence coefficients at 21 different time slots, and Figure 15f shows the pre-FEC BER of the FSO link without retransmission under the weak turbulence condition. After testing, the system can achieve error-free transmission right after FEC decoding at 5% FEC OH in the weak turbulence condition. From the above, we can see that the HARQ strategy can significantly improve the performance of the FSO/RF hybrid transmission system under the influence of atmospheric turbulence. In practice, we need to dynamically select the retransmission strategy and FEC overhead according to the planned transmitting power of the FSO and RF link, the size of turbulence, and other factors.

By substituting the actual parameters into the numerical simulation in Section 3, we summarized the output probability and throughput at different circumstances, which corresponded to the aforementioned BER performances. The SNRs at the receiving end were obtained by averaging the SNR of 21 groups of data tested under different turbulences and rate ratios in the actual experiment. These SNRs were calculated by error vector magnitude (EVM), which is called the calculated SNR.

Table 3 shows the output probability and throughput under strong turbulence intensity, with different RF to FSO data rate ratios $\Psi$, FEC overheads, and HARQ strategies. It can also shed light on the great performance improvement brought by the hybrid FSO/RF communication system with the HARQ technique.

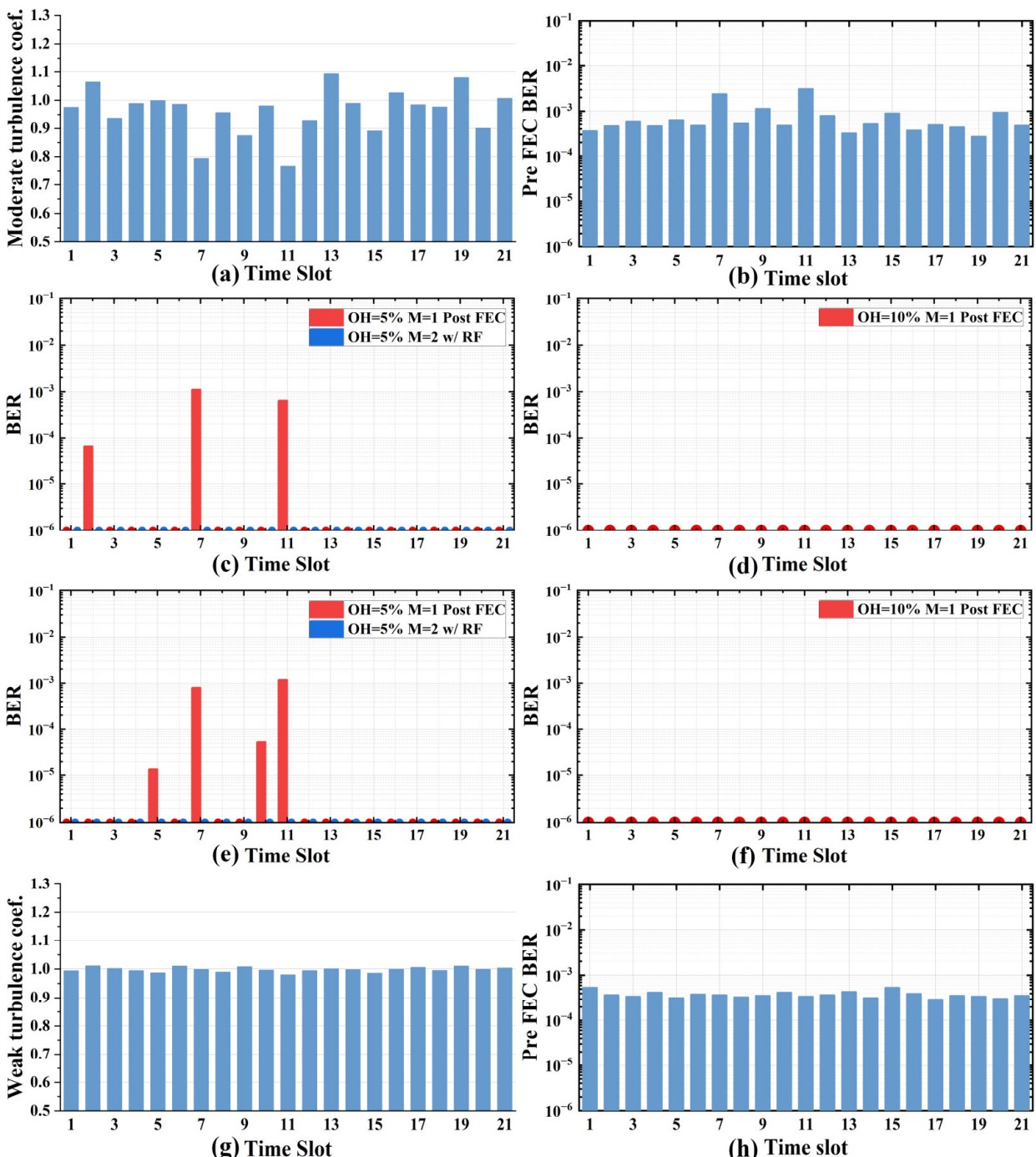

**Figure 15.** The BER performance under moderate and weak atmospheric turbulence: (**a**) moderate turbulence coefficients at 21 different time slots; (**b**) the pre-FEC BER of FSO link without retransmission under moderate atmospheric turbulence; (**c**) 5% OH with HARQ-aided BER performances at $\Psi = 0.25$ under moderate atmospheric turbulence; (**d**) 10% OH after FEC BER performance at $\Psi = 0.25$ under moderate atmospheric turbulence; (**e**) 5% OH with HARQ-aided BER performances at $\Psi = 0.5$ under moderate atmospheric turbulence; (**f**) 10% OH after FEC BER performances at $\Psi = 0.5$ under moderate atmospheric turbulence; (**g**) weak turbulence coefficients at 21 different time slots; (**h**) the pre-FEC BER of FSO link without retransmission under the weak turbulence condition.

**Table 3.** The output probability and throughput at strong turbulence intensity, with different RF to FSO data rate ratios Ψ, FEC overheads, and HARQ strategies.

| Intensity | | Strong | | | |
|---|---|---|---|---|---|
| Ψ | | 1:4 | | 1:2 | |
| FEC OH | Strategy | Calculated Outage Pr. | Calculated Throughput (bps/Hz) | Calculated Outage Pr. | Calculated Throughput (bps/Hz) |
| 5% | M = 1 | $2.33 \times 10^{-04}$ | 1.9043 | $2.33 \times 10^{-04}$ | 1.9043 |
| | M = 2 w/RF | $2.86 \times 10^{-11}$ | 1.9048 | $1.95 \times 10^{-21}$ | 1.9048 |
| | M = 3 w/RF | $3.53 \times 10^{-14}$ | 1.9048 | 0 | 1.9048 |
| | M = 4 w/RF | $3.42 \times 10^{-17}$ | 1.9048 | 0 | 1.9048 |
| | M = 2 w/RF&FSO | $2.90 \times 10^{-14}$ | 1.9043 | $1.58 \times 10^{-24}$ | 1.9043 |
| | M = 3 w/RF&FSO | $5.06 \times 10^{-19}$ | 1.9043 | 0 | 1.9043 |
| 10% | M = 1 | $1.12 \times 10^{-04}$ | 1.8180 | $1.12 \times 10^{-04}$ | 1.8180 |
| | M = 2 w/RF | $3.48 \times 10^{-12}$ | 1.8182 | 0 | 1.8182 |
| | M = 3 w/RF | $8.33 \times 10^{-16}$ | 1.8182 | 0 | 1.8182 |
| | M = 4 w/RF | $1.13 \times 10^{-20}$ | 1.8182 | 0 | 1.8182 |
| | M = 2 w/RF&FSO | $3.36 \times 10^{-15}$ | 1.8180 | 0 | 1.8180 |
| | M = 3 w/RF&FSO | $1.13 \times 10^{-20}$ | 1.8180 | 0 | 1.8180 |
| 15% | M = 1 | $5.10 \times 10^{-05}$ | 1.7390 | $5.10 \times 10^{-05}$ | 1.7390 |
| | M = 2 w/RF | $3.39 \times 10^{-13}$ | 1.7391 | 0 | 1.7391 |
| | M = 3 w/RF | $6.24 \times 10^{-18}$ | 1.7391 | 0 | 1.7391 |
| | M = 4 w/RF | 0 | 1.7391 | 0 | 1.7391 |
| | M = 2 w/RF&FSO | $3.14 \times 10^{-16}$ | 1.7390 | 0 | 1.7390 |
| | M = 3 w/RF&FSO | $8.11 \times 10^{-23}$ | 1.7390 | 0 | 1.7390 |

Note that Table 3 calculates the overall outage probability under each turbulence intensity, while the previous experiments included the worst turbulence coefficient under each turbulence intensity, so that the BER was not zero after one RF retransmission, and retransmission twice was required. In practice, the probability of the worst turbulence coefficient is consistent with the outage probability.

## 6. Conclusions

In this paper, we proposed and demonstrated a novel hybrid MMW and FSO architecture with ADCT and cc-HARQ techniques. We investigated the outage and throughput performances of the system under atmospheric turbulence influence. 10-Gbaud QPSK data were experimentally transported, and the MRC algorithm used at the receiving end fully exploited the complementary channel response of MMW and FSO links. The BER and outage probability performance improvements were tested and verified under either clean air or different turbulence intensity conditions. We provided a theoretical analysis and experimental demonstrations to validate the effectiveness of our approach. A closed-form expression for the outage probability and throughput of the RF/FSO system, considering the characteristics of atmospheric turbulence in the FSO link, was also obtained. The relevant results indicate that there is great potential for using hybrid FSO/MMW links for spatial communication in the future.

**Author Contributions:** Conceptualization, J.J. and B.D.; methodology, J.J., B.D. and J.Z.; software, Y.L., B.D. and C.H.; validation, Y.L. and B.D.; formal analysis, Y.L., B.D. and P.L.; investigation, Y.L., B.D. and P.L.; resources, J.J. and J.Z.; data curation, Y.L. and C.H.; writing—original draft, Y.L.; writing—review and editing, J.Z.; visualization, Y.L.; supervision, X.T., J.S., N.C. and J.Z. All authors have read and agreed to the published version of the manuscript.

**Funding:** This work is partially supported by National Natural Science Foundation of China (62235005, 62171137 and 61925104) and the Natural Science Foundation of Shanghai (21ZR1408700).

**Institutional Review Board Statement:** Not applicable.

**Informed Consent Statement:** Not applicable.

**Data Availability Statement:** Data underlying the results presented in this paper are not publicly available at this time but may be obtained from the authors upon reasonable request.

**Conflicts of Interest:** The authors declare no conflict of interest.

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
