# Peer review of "A Hybrid Millimeter-Wave and Free-Space-Optics Communication Architecture with Adaptive Diversity Combining and HARQ Techniques"

_photonics, doi:10.3390/photonics10121320_

Round 1

Reviewer 1 Report

A hybrid MMW/FSO communication system with ADCT and cc-HARQ techniques (including RF-only retransmission or RF/FSO retransmission strategies) is discussed in this manuscript. The manuscript is well organized and well written. Here are some suggestions to make it more comprehensive.

1.      Atmospheric turbulence induced fading coefficients are added to the data stream at the transmitter. Generally, channel fading is reflected in flickering of received optical power. It would be better to discuss the rationality of adding turbulence fading coefficient at the transmitter.

2.      ARQ feedback also brings time delay. The more retransmission times, the farther the distance, and the more time it takes. When considering throughput, what would happen if the time consumed by ARQ is included.

Reviewer 2 Report

This paper proposes a hybrid FSO/RF communication architecture designed for enhancing satellite communications. The contribution of this work remains unclear due to the absence of both a comparative analysis and references to preceding solutions.

- The introductory section lacks sufficient detail, failing to adequately present both the problem addressed within this paper and the background efforts undertaken in this field. It is recommended to restructure the discussion of related works, starting from earlier solutions and progressing to the present developments. In the analysis of previous works, emphasis should be placed on mentioning the main contributions, as well as the advantages and disadvantages of each individual work, which are currently absent.

- List the values of the parameters you used in the simulation in a table. Are these values practical? More references should be cited.

- The same applies to the experiment. The tabulation of basic quantities' values is advisable, along with a contextualization of the experiment's scale. 

- How is the proposed architecture affected by weather effects that affect RF and FSO links differently?

- The quality of the schemes and table is low and needs to be improved. Notably, Figure 9 appears to be perplexing for readers and should be revised for greater clarity.

-

Reviewer 3 Report

The paper titled “A hybrid millimeter-wave and free-space-optics communication architecture with adaptive diversity combining and HARQ techniques” focuses on a novel MMW and FSO hybrid architecture, which is an important topic of satellite communication. The paper is well-written, with comprehensive theoretical study and solid experimental demonstrations. The paper can be accepted after some minor revisions.

1.     Many abbreviations must be defined before using them, even in the abstract session, for example, RF/FSO, QPSK, FEC, etc. Please correct them and double-check the whole paper.

2.     What exactly the functions should be included in the HARQ Controller in Fig. 2. Please add more analysis.

3.     Please describe more how turbulence coefficients are generated and added to the channel in the experiment. What model is used for the turbulence channel in your experiment?

4.     The wireless transmission distance is around 4m. Are there any limitations to your setup? What is the gain of your antennas?

5.     Please add more description of your MMW and FSO system setups, for example, what gain of PA or LNA is used in your system, what is the receiving power of the RF link, and what aperture of your FSO antennas and so on.

Reviewer 4 Report

This manuscript proposes a novel communication system that combines millimeter-wave and free-space-optics technologies, using adaptive diversity combining and hybrid automatic repeat request techniques to improve data transmission speed, reliability, and throughput under various atmospheric turbulence conditions. However, the following concerns need to be addressed before publication consideration:

1. “Thus, each transmission and retransmission time slot of the FSO link is 10^3 times larger than the turbulence frequency which is millisecond order”. Here, the authors want to express that the duration of each FSO symbol is much longer than the period of turbulence variation, so the effect of turbulence within each symbol can be ignored. However, the authors use the frequency of turbulence variation to indicate the period of turbulence variation, which may cause some confusion.

2. In the numerical simulation section, further clarification is needed on the specific definition of M. According to the definition in previous papers, M represents the number of retransmissions. In this case, what is the relationship between M and M_RF, M_FSO in different RF/FSO hybrid transmission modes in the legend of this section?

3. In Figure 6(a), does the data for M=3w/RF overlap with the data for M=2w/RF&FSO? Similarly, in Figure 7(a), does the data for M=3w/RF overlap with the data for M=4w/RF? If so, it is suggested to use different point types to present the relationship of overlapping data.

4. Why there is no data comparison for M=4w/ RF&FSO in Figures 6 and 7 of the numerical simulation section?

The grammer of this paper is correct and the writing is fluent.

Reviewer 5 Report

The authors presented a very interesting concept and did justice in formulating the theory and experimental verification. It is important for the authors to split Figure 9. It looks too congested in its current form. The Authors should also add a photo of the actual experimental setup for validation. The Authors are also not consistent in the use of markers in the plots. Some have different marker styles while others don't. The quality of the plots and layouts must be significantly improved, currently not appealing. Use different lines/markers in Figures 6, 7, and 8. Many of your references are old. Look for more recent and relevant published literature!

Well-written article overall. However, minor grammatical errors in the paper must be corrected, especially at the beginning section of the paper. 

Round 2

Reviewer 2 Report

No further comments. Minor editing of the text is required.

N/A